# Factors associated with unsuppressed viremia in women living with HIV on lifelong ART in the multi-country US-PEPFAR PROMOTE study: A cross-sectional analysis

Patience Atuhaire[1]☯*, Sherika Hanley[2]☯, Nonhlanhla Yende-Zuma[3], Jim Aizire[4], Lynda Stranix-Chibanda[5], Bonus Makanani[6], Beteniko Milala[7], Haseena Cassim[8], Taha Taha[4], Mary Glenn Fowler[9]

**1** Makerere University-Johns Hopkins University (MU-JHU) Kampala, Uganda, **2** Centre for the AIDS Programme of Research in South Africa (CAPRISA), Umlazi Clinical Research Site, Nelson R. Mandela School of Medicine, Durban, South Africa, **3** Centre for the AIDS Programme of Research in South Africa (CAPRISA), Durban, South Africa, **4** Johns Hopkins Bloomberg School of Public Health, Department of Epidemiology, Baltimore, MD, United States of America, **5** University of Zimbabwe College of Health Sciences Department of Paediatrics and Child Health, Harare, Zimbabwe, **6** Malawi College of Medicine-John's Hopkins Research Project, Blantyre, Malawi, **7** University of North Carolina (UNC) Project, Lilongwe, Malawi, **8** Perinatal HIV Research Unit (PHRU), Chris Hani Baragwanath Hospital, University of Witwatersrand, Johannesburg, South Africa, **9** Johns Hopkins University, Departments of Pathology and Epidemiology, Baltimore, MD, United States of America

☯ These authors contributed equally to this work.
* patuhaire@mujhu.org

## Abstract

### Background

Despite recent efforts to scale-up lifelong combination antiretroviral therapy (cART) in sub-Saharan Africa, high rates of unsuppressed viremia persist among cART users, and many countries in the region fall short of the UNAIDS 2020 target to have 90% virally suppressed. We sought to determine the factors associated with unsuppressed viremia (defined for the purpose of this study as >200 copies/ml) among sub-Saharan African women on lifelong cART.

### Methods

This cross-sectional analysis was based on baseline data of the PROMOTE longitudinal cohort study at 8 sites in Uganda, Malawi, Zimbabwe and South Africa. The study enrolled 1987 women living with HIV who initiated lifelong cART at least 1–5 years ago. Socio-demographic, clinical, and cART adherence data were collected. We used multivariable Poisson regression with robust variance to identify factors associated with unsuppressed viremia.

### Results

At enrolment, 1947/1987 (98%) women reported taking cART. Of these, HIV-1 remained detectable in 293/1934 (15%), while 216/1934 (11.2%) were considered unsuppressed (>200 copies/ml). The following factors were associated with an increased risk of

**Data Availability Statement:** All relevant data are within the paper and its Supporting Information files.

**Funding:** The PROMOTE study is funded by the President's Emergency Plan for AIDS Relief (PEPFAR) through DAIDS/NIAID/NIH grants to each of the following Clinical Trials Units (CTUs): JHU-Uganda CTU Makerere University-Johns Hopkins University (MU-JHU) Research Collaboration, grant # UM1 AI069530-11; The Johns Hopkins University-Blantyre Clinical Trials Unit, grant # UM1AI069518-12; The University of North Carolina Global HIV Prevention and Treatment Clinical Trials Unit, grant # 5UM1AI069423-12; University of Zimbabwe College of Health Sciences Clinical Trials Research Centre, grant # 5UM1AI069436-12; PHRU KARABELO Clinical Trials Unit for NIAID Networks Grant # 5UM1AI069453; Clinical Trials Unit for AIDS/Tuberculosis Prevention and Treatment - Grant Number: 5UM1AI069469-11; and CAPRISA Clinical Trials Unit for AIDS/Tuberculosis Prevention and Treatment, grant # 5UM1AI069469. The funders had no role in study design, data collection and analysis, decision to publish, or preparation of the manuscript.

**Competing interests:** The authors have declared that no competing interests exist.

unsuppressed viremia: not having household electricity (adjusted prevalence risk ratio (aPRR) 1.74, 95% confidence interval (CI) 1.28–2.36, p<0.001); not being married (aPRR 1.32, 95% CI 0.99–1.78, p = 0.061), self-reported missed cART doses (aPRR 1.63, 95% CI 1.24–2.13, p<0.001); recent hospitalization (aPRR 2.48, 95% CI 1.28–4.80, p = 0.007) and experiencing abnormal vaginal discharge in the last three months (aPRR 1.88; 95% CI 1.16–3.04, p = 0.010). Longer time on cART (aPRR 0.75, 95% CI 0.64–0.88, p<0.001) and being older (aPRR 0.77, 95% CI 0.76–0.88, p<0.001) were associated with reduced risk of unsuppressed viremia.

## Conclusion

Socioeconomic barriers such as poverty, and individual barriers like not being married, young age, and self-reported missed doses are key predictors of unsuppressed viremia. Targeted interventions are needed to improve cART adherence among women living with HIV with this risk factor profile.

## Introduction

Since 2012, the rapid scale up of the World Health Organization (WHO) option B+ strategy among pregnant or breastfeeding women living with Human Immunodeficiency Virus (HIV) has resulted in a substantial reduction in maternal morbidity and mortality, as well as incident pediatric HIV infections[1]. Subsequently with the introduction of the Universal "Test and Treat" strategy, approximately 23.3 million people (including women) had access to combination Antiretroviral Therapy (cART) globally in 2018 [2]. Ensuring sustained adherence to and virologic suppression on cART is paramount in achieving the Joint United Nations Programme on HIV and AIDS (UNAIDS) 90-90-90 2020 strategy in ending the epidemic by 2030 [3–5].

Barriers to achieving the UNAIDS 2020 strategy regarding the 'third 90' persist in sub-Saharan Africa in part due to suboptimal cART adherence [4, 5]. In the absence of viral resistance, HIV viral load assessment is the proxy for adherence, therefore the contributing factors to both adherence and viremia may overlap. The most common factors identified as being associated with decreased adherence include individual factors like younger age below 24 years, forgetting the dosing time, depression, and substance use[6, 7]. The predominant contextual issue remains stigmatization and disclosure [4–7]. Other factors such as length of time on ART, level of education, personal motivation to start ART and satisfaction with health worker information availed [6–8]. Additionally, studies have shown that adherence to cART is better during pregnancy compared to the post-partum period [4, 8]. This may be attributed to less motivation to take ART since the risk of HIV transmission has ceased after cessation of breast feeding, as well as a possible break in transition from postnatal to general HIV care [4, 5].

Whereas the scale up of virologic monitoring in sub-Saharan Africa since 2013 has led to the availability of data regarding factors associated with virologic detectability, there is a paucity of literature as to which factors are most strongly associated with unsuppressed viremia among African pregnant women and mothers living with HIV[9]. What is known to date is that virological detectabilty in resource-limited settings has been associated with the presence of comorbidities like tuberculosis or psychiatric disease, higher pretreatment HIV RNA levels, repeat testers after suspected virologic failure and initiation of cART late in pregnancy[6, 7,

10–13]. The vast majority of existing studies have assessed factors associated with viremia >1000 or >400 copies/ml [6, 7, 10, 11]. Emerging antiretroviral drug resistance has been known to occur from levels of 200 copies/ml or above, and use of this threshold eliminates most cases of apparent viremia caused by viral load blips or assay variability[14, 15].Therefore this analyses sought to determine clinical and demographic risk factors associated with unsuppressed viremia above 200 copies/ml among a well characterized cohort of women living with HIV originally in the IMPAACT 1077BF/1077FF PROMISE (Promoting Maternal and Infant Survival Everywhere) clinical trial at the time of their entry into the PROMOTE Study. This cohort presents a unique opportunity to assess longer term treatment outcomes among women previously randomized to different antiretroviral (ARV) regimens (WHO option A and option B) during pregnancy and postpartum for the purpose of preventing perinatal HIV transmission[16]. Subsequently these women transitioned to lifelong cART in response to the START study which showed clear benefit of universal ART in June 2015(10). The PROMOTE study approach provides data from current public sector HIV care provision mixed with precise individualized clinical and laboratory data collected under trial settings.

## Materials and methods

### Design

The PEPFAR-PROMOTE study is a five-year observational cohort of sub-Saharan African women with HIV and their children previously enrolled in the PROMISE trial for the prevention of mother to child transmission of HIV [17]. The study is being conducted at eight research sites in four African countries: MUJHU/Kampala (Uganda), Blantyre and Lilongwe (Malawi), Harare Family Care, Seke North and St. Mary's (Zimbabwe), PHRU/Johannesburg and CAPRISA Umlazi/Durban (South Africa). Commencing three months after the PROMISE trial closed-out, 1987 mothers and their children were recruited from September 2016 to August 2017 into the PROMOTE study. A cross-sectional analysis of unsuppressed viremia and related factors was conducted at enrolment.

### Setting and study populations

At entry into the PROMISE trial, these women were pregnant, asymptomatic with pre-ART CD4+ T-cell counts ≥ 350 cells/mm3 and more details of eligibility criteria into the PROMISE trial have been previously published[16]. At enrollment into the PROMOTE study, women were either having a subsequent pregnancy or not pregnant (up to 4 years post-delivery) or breastfeeding babies born between PROMISE study close up and PROMOTE study start (Fig 1). Uganda site had women on both LPV/r (69%) and EFV (29.8%) based regimens, while Zimbabwe and Malawi had most women (97.3% and 98.3% respectively) on EFV based regimens only and South Africa had 96.6% of the women on the fixed dose combination (TDF/FTC/EFV).

### Inclusion criteria

Women and children enrolled in the PROMISE trial from the 8 African PROMISE sites described above, who were willing to provide informed consent to enroll and continue follow-up in the PROMOTE study.

### Exclusion criteria

Women who were judged by the site team as having social or other reasons which would make it difficult for the mother/child pair to comply with study requirements, for example those

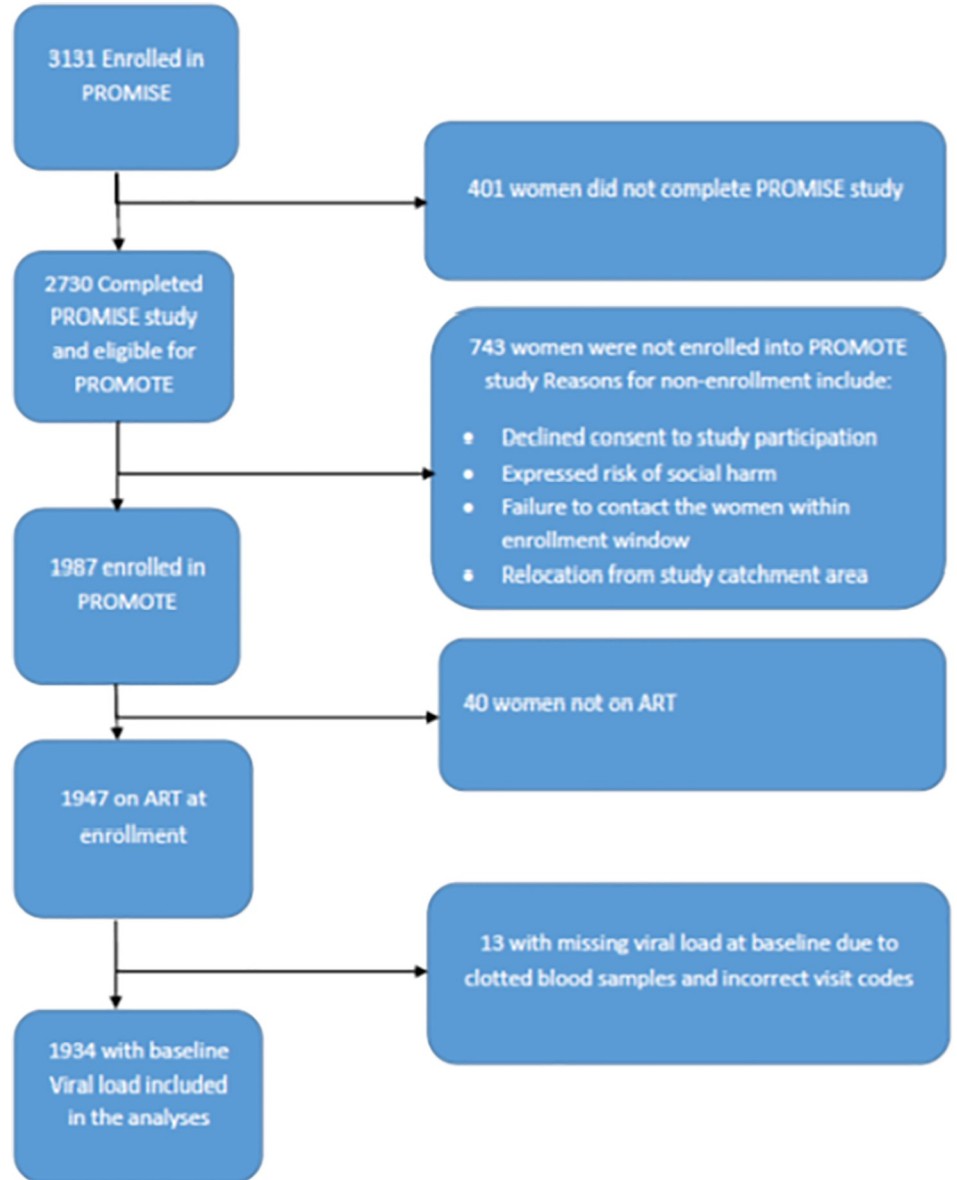

**Fig 1. Flow chart showing numbers from PROMISE trial to PROMOTE study.**

women who felt that continued follow up may predispose them to social harm and women who planned on relocating home out of the study catchment area and could not keep up with study visit requirements afterwards were excluded from the study.

### Enrolment study procedures for women

Mothers were enrolled after appropriate counseling and consenting in the PROMOTE study. At the enrolment visit, trained study workers administered socio-demographic and ART adherence questionnaires. A complete medical history and physical examination, including WHO clinical staging, was performed. Included in a holistic package of comprehensive counseling was the provision of study-specific antiretroviral adherence counseling. Enrollment laboratory evaluations included: viral load and CD4+ cell count. Assays were done at various

Good Clinical Laboratory Practice (GCLP) and Clinical Laboratory Improvement Amendments (CLIA) compliant site laboratories adherent to standards for proper collection, processing, labeling, and transport of specimens. The COBAS TaqMan and Abbot viral load assays were used with a lower limit of detectability at 20 copies/ml and 40 copies/ml respectively. All questionnaire responses and laboratory data were completed on designated case report forms (CRFs) by trained research site personnel.

## Ethical considerations

The PROMOTE study was approved by all relevant institutional review boards (IRBs) in the U. S. and participating African research sites/countries. These include MUJHU/Kampala, Uganda: The Joint Clinical Research Centre (JCRC) IRB in Uganda and The Johns Hopkins Medical Institutions (JHMI) IRB in the U.S.; Umlazi, Durban, South Africa: Biomedical Research Ethics Committee and Kwazulu-Natal Department of Health; Harare, Seke North and St. Mary's sites, Zimbabwe: Medical Research. Council of Zimbabwe (MRCZ) National Ethics Committee; Blantyre, Malawi: College of Medicine Research and Ethics Committee (COMREC) in Malawi and Johns Hopkins Medical Institutions (JHMI) IRB in the U.S.; Lilongwe, Malawi: National Health Sciences Research Committee (NHSRC) in Malawi and University of North Carolina, Chapel Hill (UNC-CH) Office of Human Research Ethics IRB in the U.S.; and PHRU, Johannesburg, South Africa: University of Witwatersrand Human Research Ethics Committee (Medical). All women provided written informed consent to enroll and be followed up for the duration of the study with their children and agreed to provide study samples for protocol lab safety assays, as well as for storage of blood and hair samples.

## Statistical analysis

We analyzed baseline data from a multi-country cohort study to estimate the proportion of women living with HIV who had unsuppressed viremia (defined as viral load above 200 copies/ml) and to identify predictors of unsuppressed viremia. Furthermore, in accordance with WHO threshold for treatment failure, we performed sensitivity analyses and defined unsuppressed viremia as viral load above 1000 copies/ml. Fisher's exact, Chi-square test of independence or Wilcoxon rank sum tests were used to test for an association between baseline characteristics and unsuppressed viremia. We used univariable and multivariable Poisson regression models with robust variance to identify the predictors of unsuppressed viremia, and calculated prevalence risk ratios to measure the strength of an association between baseline characteristics and unsuppressed viremia. This was done in two ways (i) each predictor was fitted in the model, and (ii) multivariable model with multiple predictors included in the model. Variables included in the multivariable analyses were chosen based on prior research on risk factors, clinical associations and biological plausibility. Due to site country variability in ARV regimen type, we did not adjust for the ARV regimen in any of the models to avoid multi-collinearity between country and ARV regimen variables. Variables with increased missing data (>20% of observations missing per variable) and variables that were highly correlated were not included in the multivariable model. In an exploratory analyses, women with detectable viral loads were further stratified into the following thresholds (<50, 50–200, 201–1000 and >1000 copies per ml based on the varied thresholds by various HIV cART committees in resource rich and resource limited settings)[18, 19].

## Results

Overall, 1987 mothers were enrolled into the PROMOTE study, of whom 1947 (98%) women reported taking ART at the enrolment visit and HIV-1 viral load results were available for 1934.

HIV-1 VL was above the limit of quantification in 293/1934 (15%). A total of 216/1934 (11.2%) presented with an unsuppressed viremia above 200 copies/ml. Furthermore, among the 293 women with detectable viral load, 24 (8.2%) had VL below 50, 53 (18.1%) had VL between 50 and 200, 50 (17.1%) had VL between 201 and 1000, while 166 (56.7%) had VL above 1000 copies/ml.

The individual and contextual baseline characteristics of the PROMOTE study have been reported elsewhere [17]. Table 1 displays baseline characteristics stratified by viral load below

**Table 1. Individual and contextual baseline characteristics.**

| Variable | Viral load ≤200 copies/ml (N = 1718) | Viral load >200 copies/ml (N = 216) | Total | p-value |
|---|---|---|---|---|
| **Socioeconomic and demographic factors** | | | | |
| *Country, n (%)* | | | | <0.001 |
| Uganda | 319 (90.6%) | 33 (9.4%) | 352 | |
| Malawi | 522 (82.5%) | 111 (17.5%) | 633 | |
| Zimbabwe | 406 (90.6%) | 42 (9.4%) | 448 | |
| South Africa | 471 (94.0%) | 30 (6.0%) | 501 | |
| Age (years), median(IQR) | 31 (28–35) | 29 (25–33) | 31(27–35) | <0.001 |
| Baseline CD4 cell count (cells/µL),median (IQR) | 852 (674–1064) | 615 (472–810) | 825 (646–1040) | <0.001 |
| *Marital status, n(%)* | | | | 0.003 |
| Other (single, divorced, widowed, separated) | 325 (84.4%) | 60 (15.6%) | 385 | |
| Married/regular partner | 1393 (89.9%) | 156 (10.1%) | 1549 | |
| *Employment, n(%)[a]* | | | | 0.076 |
| Formal employment | 383 (90.5%) | 40 (9.5%) | 423 | |
| Self-employment (small business) | 532 (86.5%) | 83 (13.5%) | 615 | |
| Not employed/housewife | 802 (89.7%) | 92 (10.3%) | 894 | |
| *Highest level of education, n(%)* | | | | 0.011 |
| Secondary school completed or tertiary education | 1227 (90.0%) | 136 (10.0%) | 1363 | |
| Lower level of education | 491 (86.0%) | 80 (14.0%) | 571 | |
| *Electricity in the premises, n(%)* | | | | |
| Available | 1209 (91.8%) | 108 (8.2%) | 1317 | <0.001 |
| Not available | 509 (82.5%) | 108 (17.5%) | 617 | |
| *Tap water in the premises, n(%)* | | | | |
| Available | 1139 (90.4%) | 121 (9.6%) | 1260 | 0.004 |
| Not available | 579 (85.9%) | 95 (14.1%) | 674 | |
| *Travel time from home to clinic, n(%)[b]* | | | | 0.044 |
| Less than 30 minutes | 444 (92.3%) | 37 (7.7%) | 481 | |
| 30–60 minutes | 757 (87.7%) | 106 (12.3%) | 863 | |
| 1–2 hours | 389 (87.2%) | 57 (12.8%) | 446 | |
| Greater than 2 hours | 127 (88.8%) | 16 (11.2%) | 143 | |
| *Disclosed HIV status to partner, n(%)[c]* | | | | 0.465 |
| Disclosed to partner | 1201 (90.2%) | 131 (9.8%) | 1332 | |
| No disclosure | 192 (88.5%) | 25 (11.5%) | 217 | |
| *Partner's HIV status[d]* | | | | |
| Positive | 724 (90.2%) | 79 (9.8%) | 803 | 0.908 |
| Negative | 257 (89.9%) | 29 (10.1%) | 286 | |
| *Condom usage during sex in last 3 months, n(%)[e]* | | | | 0.811 |
| Always | 502 (89.8%) | 57 (10.2%) | 559 | |
| Sometimes | 506 (88.6%) | 65 (11.4%) | 571 | |
| Never | 265 (88.9%) | 33 (11.1%) | 298 | |
| *Years on ART, median(IQR)* | 2 (1–2) | 1 (1–2) | 2 (1–2) | <0.001 |
| **Clinical factors** | | | | |

*(Continued)*

**Table 1.** (Continued)

| Variable | Viral load ≤200 copies/ml (N = 1718) | Viral load >200 copies/ml (N = 216) | Total | p-value |
|---|---|---|---|---|
| *Admitted to hospital in the past 3 months, n(%)* | | | | |
| Yes | 22 (75.9%) | 7 (24.1%) | 29 | 0.036 |
| No | 1696 (89.0%) | 209 (11.0%) | 1905 | |
| *Received TB treatment in the last 3 months, n(%)* | | | | |
| Yes | 8 (88.9%) | 1 (11.1%) | 9 | 1.00 |
| No | 1710 (88.8%) | 215 (11.2%) | 1925 | |
| *Presence of abnormal vaginal discharge in the last 3 months, n (%)* | | | | |
| Yes | 80 (82.5%) | 17 (17.5%) | 97 | 0.047 |
| No | 1638 (89.2%) | 199 (10.8%) | 1837 | |
| *Currently breastfeeding, n(%)[b]* | | | | |
| Yes | 163 (84.9%) | 29 (15.1%) | 192 | 0.071 |
| No | 1554 (89.3%) | 187 (10.7%) | 1741 | |
| **ART related factors** | | | | |
| *ART regimen* | | | | |
| EFV or NVP based[f] | 1007 (86.4%) | 158 (13.6%) | 1165 | <0.001 |
| LPV/r based[g] | 252 (90.0%) | 28 (10.0%) | 280 | |
| Fixed dose combination (TDF/FTC/EFV) | 459 (93.9%) | 30 (6.1%) | 489 | |
| *Since last scheduled visit, when was the last time you missed any of your ARV doses? n(%)[h]* | | | | |
| Never missed any doses | 1257 (91.8%) | 113 (8.2%) | 1370 | <0.001 |
| Within the last week | 72 (79.1%) | 19 (20.9%) | 91 | |
| 1–2 weeks ago | 40 (85.1%) | 7 (14.9%) | 47 | |
| 2–4 weeks ago | 64 (83.1%) | 13 (16.9%) | 77 | |
| 1–3 months ago | 183 (85.9%) | 30 (14.1%) | 213 | |
| More than 3 months ago | 61 (77.2%) | 18 (22.8%) | 79 | |
| Don't know or cannot remember | 33 (91.7%) | 3 (8.3%) | 36 | |
| *Number of days ARV doses missed in last four days, n(%)[h]* | | | | |
| None | 1638 (90.6%) | 169 (9.4%) | 1807 | <0.001 |
| One day | 52 (86.7%) | 8 (13.3%) | 60 | |
| Two days | 8 (66.7%) | 4 (33.3%) | 12 | |
| Three days | 1 (50.0%) | 1 (50.0%) | 2 | |
| Four days | 9 (34.6%) | 17 (65.4%) | 26 | |
| Don't know or cannot remember | 2 (33.3%) | 4 (66.7%) | 6 | |
| *Awareness of dosing instructions, n(%)[h]* | | | | |
| Yes | 826 (87.9%) | 114 (12.1%) | 940 | 0.017 |
| No | 842 (90.3%) | 90 (9.7%) | 932 | |
| Don't know or cannot remember | 42 (100.0%) | 0 | 42 | |

[a]2 participants had missing data

[b]1 participant had missing data

[c]amongst 1549 women with partners

[d]amongst women whom their partner's got tested and women knew their HIV status

[e]amongst women who report sexual activities, EFV/NVP based[f] (ABC/3TC or AZT/3TC or TDF/3TC), LPV/r based[g] (ABC/3TC or AZT/3TC or TDF/3TC or TDF/FTC)

[h]amongst those who were on ART at enrollment. Please note that we presented row percentages to display the proportion of patients with detectable viremia for each level of the variable.

and above 200 copies/ml. With the exception of employment status, HIV status disclosure, condom usage, all the baseline variables were associated with unsuppressed viremia >200 copies/ml (Table 1). Notably, the prevalence of unsuppressed viremia > 200 copies/ml was the

**Table 2. Factors associated with detectable viral load >200 copies/ml.**

| Variable | Multivariable[1] | | Multivariable[2] | |
|---|---|---|---|---|
| | RR (95% CI) | p-value | aRR (95% CI) | p-value |
| Age (5-year increase) | 0.77 (0.67–0.88) | <0.001 | 0.77 (0.67–0.88) | <0.001 |
| Marital status (ref: married/regular partner) | | | | |
| Other (single, divorced, widowed, separated) | 1.55 (1.19–2.04) | 0.001 | 1.32 (0.99–1.78) | 0.061 |
| Employment (ref: formal employment) | | | | |
| Not employed | 0.92 (0.65–1.32) | 0.655 | 0.84 (0.58–1.21) | 0.349 |
| Self employed | 1.09 (0.75–1.58) | 0.646 | 1.11 (0.76–1.63) | 0.577 |
| Education (ref: secondary school not complete) | | | | |
| Secondary school complete | 1.04 (0.78–1.38) | 0.806 | 1.17 (0.85–1.61) | 0.326 |
| Electricity in the premises (ref: Yes) | | | | |
| No | 1.62 (1.22–2.14) | <0.001 | 1.74 (1.28–2.36) | <0.001 |
| Tap water in the premises (ref: Yes) | | | | |
| No | 1.24 (0.96–1.62) | 0.103 | - | - |
| Travel time to clinic from home (ref: less than 1 hour) | | | | |
| 1 hour or more | 1.01 (0.77–1.33) | 0.936 | 0.88 (0.66–1.19) | 0.417 |
| Condom use during sex in last three months (ref: Never) | | | | |
| Always | 1.12 (0.74–1.72) | 0.585 | - | - |
| Sometimes | 1.05 (0.71–1.56) | 0.811 | - | - |
| HIV status disclosure to partner (ref = Yes) | | | | |
| No | 1.45 (0.94–2.25) | 0.096 | - | - |
| Abnormal vaginal discharge in last three months (ref: No) | | | | |
| Yes | 2.12 (1.32–3.39) | 0.002 | 1.88 (1.16–3.04) | 0.010 |
| Hospital admission in last three months (ref: No) | | | | |
| Yes | 2.29 (1.20–4.36) | 0.012 | 2.48 (1.28–4.80) | 0.007 |
| Aware of ARV medication dosing instructions (ref: No) | | | | |
| Yes | 1.05 (0.79–1.41) | 0.723 | 1.08 (0.81–1.43) | 0.612 |
| Missed ART doses since last visit (ref: None) | | | | |
| Missed some doses | 2.01 (1.55–2.60) | <0.001 | 1.63 (1.24–2.13) | <0.001 |
| Time since ART initiation (per 1-year increase) | 0.68 (0.57–0.81) | <0.001 | 0.75 (0.64–0.88) | <0.001 |

[1]Each predictor fitted separately while adjusted for the country
[2]Multivariable model with multiple predictors

highest (17.5%) in Malawi compared to other countries. Additionally, 24% of women with recent hospitalization had unsuppressed viremia.

Results from the multivariable model are shown in Table 2. Recent hospital admission and experiencing abnormal vaginal discharge in the last three months were associated with a 2.5-fold and almost 2-fold higher risk of detectable viremia respectively (adjusted prevalence risk ratio (aPRR) 2.48, 95% confidence interval (CI) 1.28–4.80, p = 0.007; aPRR 1.88; 95% CI 1.16–3.04, p = 0.010). In addition, the absence of socioeconomic factors such as electricity in the household premises was associated with a 74% higher risk of detectable viremia (aPRR 1.74, 95% CI 1.28–2.36, p<0.001). Married or women with regular partner had a 32% increased risk of presenting with detectable viremia (aPRR 1.32, 95% CI 0.99–1.78, p = 0.061) compared to women who were neither married nor have partners. Women who either missed some of their ART doses or cannot recall missing any doses since the last scheduled visit were more likely to present with detectable viremia (aPRR 1.63, 95% CI 1.24–2.13, p<0.001)

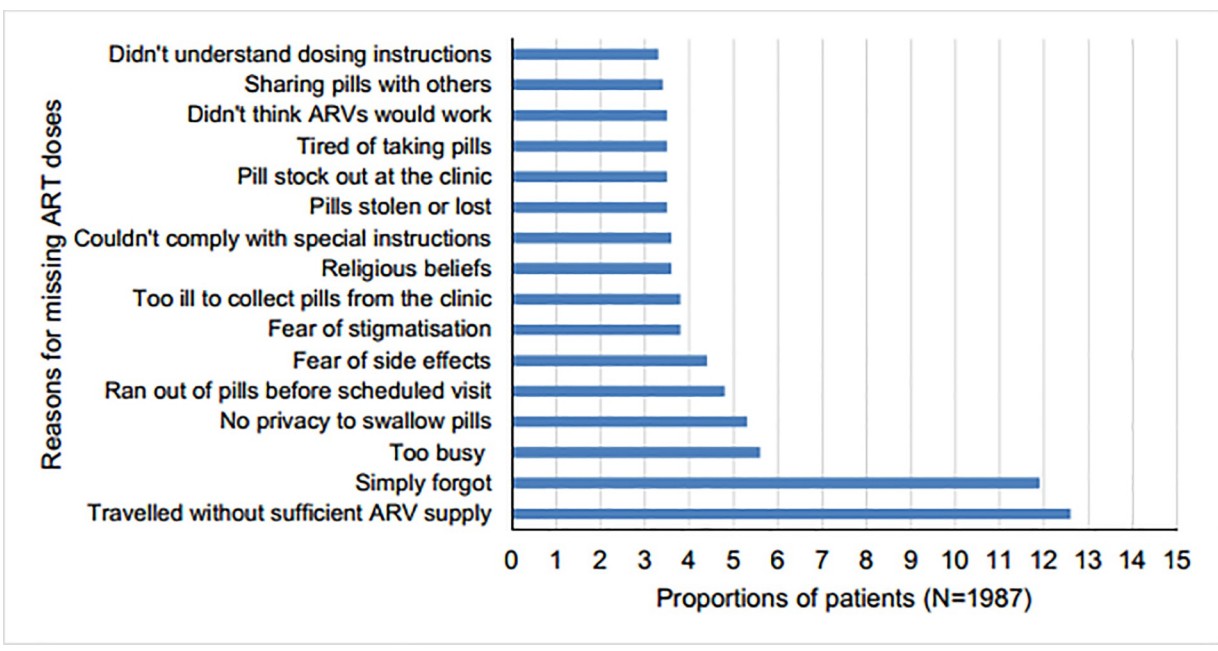

**Fig 2. Reasons for missing ART dose.**

compared to women who never missed any doses. Longer exposure to ART (aPRR: 0.75, 95% CI 0.64–0.88), p<0.001) and being older (aPRR 0.77, 95% CI 0.76–0.88, p<0.001) were associated with lower risk of detectable viremia. Other variables that were tested did not show strong association with detectable viremia such as: secondary school level completion or tertiary education attainment (aPRR 1.17, 95% CI 0.85–1.61, p = 0.326; compared to women with lower level of education); travel time to the cART clinic of 1 hour or more reduces the risk of detectable viremia by 12% when compared to travelling for less than an hour (aPRR 0.88, 95% CI 0.66–1.91, p = 0.417) and being aware of antiretroviral (ARV) medication dosing instructions (aPRR 1.08, 0.81–1.43, p = 0.612).

Notably, the most common reason for missing ART dosing was travelling without sufficient ARV supply and simply forgetting (Fig 2). Despite 11% of women not disclosing their HIV status to their male partners, this variable was not significantly associated with detectable viremia.

Results of the sensitivity analyses using the WHO threshold of 1000 copies/ml did not show any major differences in the predictors of unsuppressed viremia (S3 Table).

## Discussion

We found that 89% of the 1934 women initiated on antiretroviral treatment had suppressed viremia ≤200 copies/ml. This is slightly below the UNAIDS 90% target bearing in mind that the 200 copies/ml threshold is lower than the 1000 copies threshold used by UNAIDS. Among the women with detectable viremia, we identified that socio demographic, self-reported non-adherence, and clinical factors were associated with detectable viremia > 200copies/ml.

The higher proportion of women with detectable viremia than the UNAIDS target suggests that challenges to achieving the 3rd '90' still persist even among women who are in an ideal setting (research) compared to the programs in resource limited settings[5, 9]. The association between socio demographic factors namely the absence of household electricity, a proxy for lower economic status, and detectable viremia above 200 copies/ml, were consistent with other literature that has highlighted that socio demographic factors are key predictors to poor

adherence despite one being on cART [7, 20]. The association between younger age and unsuppressed viremia is consistent with current literature [5, 6]. Additionally, longer duration of cART use as protective of unsuppressed viremia is consistent with other literature as well [21]. In contrast to other literature, we found that employment, education and travel time to the clinic were not predictive of unsuppressed viremia in the PROMOTE cohort at baseline in the multivariable analyses [4, 20].

Based on self-reported adherence reports using the study questionnaire, we found that missed ART doses was significantly associated with unsuppressed viremia. Even though there is uncertainty regarding the reliability of self-reported adherence, this measure of adherence still remains as a cheap and easily determined mode of adherence monitoring in resource limited settings using appropriate tools[7]. We noted however that a small proportion of women (9.7%) were not aware of the dosing instructions. This factor was not a significant predictor of viral detectability. This finding may suggest that patient education is still lacking. Effective interventions like motivational adherence counseling ensure two-way input and steers away from the traditional methods of adherence counselling[5].

The most common reason for missed doses was travelling with insufficient ARV supply and forgetting, which are in line with other literature[20]. Re-emphasis on simple measures for example, setting an alarm (e.g. mobile phone alarm) and linking dosing with daily activities, should be part and parcel of adherence counseling. Provision of a pillbox is a tool used by participating South African sites. Traveling without ARV supply as a reason could fall in the "forgetting" category, or could be a cover for non-disclosure while visiting family homes. The reason of no privacy, be it at work or in home, also implies non -disclosure and feared stigmatization. Other reasons included being too busy and running out of treatment prior to visit. This is commonly due to life's every day demands including work commitments. Countries are now working towards improving the access to medicines by means of decentralized dispensing for convenient pill collection. South Africa has implemented a new model where the dispensing services are contracted to private pharmacies[22]. Other means of differentiated care strategies to ensure patient convenience include multi-month prescriptions, fast-track refills and community adherence groups who assist with collection and distribution of cART as done in Malawi, Uganda and Zimbabwe[5, 23].

Relatedly, 11% of all the women had not disclosed their HIV status to their primary partner but this was not predictive of detectable viremia. This PROMOTE result is contrary to results of other studies citing non-disclosure as being associated with poor adherence in different HIV infected populations [20, 21]. Contextual issues however remain a major underlying contributor to detectable viremia and patients may not be forthcoming with these reasons for a missed dose. Challenges in the efforts to combat stigma related issues are persistent in hyperendemic settings even with advanced HIV outreach programmes with effective adherence promotion programmes.

Whereas adherence barriers like fear of side effects, pill burden, perceptions that ART is harmful, feeling sick and depressed have been associated with virologic detectability in various AIDS Clinical Trial Participants in United States, current first-line cART comprises low pill burden and improved safety profile with low toxicity[24]. Even if about 89% of women were receiving Efavirenz or Neviparine based regimen, ART regimen type was not significantly associated with virologic detectability. The aim of the health system and its providers is to ensure adherence to first-line cART regimens to prevent the need for second- and third-line cART which are more toxic and have greater pill burden.

More so, recent hospitalization was significantly associated with viral detectability. Recent hospitalization may suggest clinical failure in this subgroup of non-suppressed women. Hospital readmissions are frequent among HIV positive adults when compared to their HIV

negative counterparts [25]. Persistent viremia above 200 copies/ml has been found to predispose one to higher morbidity, virologic failure and mortality especially among cases of delayed ART switch to 2nd line therapy [15, 26].

Recent abnormal vaginal discharge was also significantly associated with unsuppressed viremia. Vaginal discharge may imply Sexual Transmitted Infections (STIs) in these women. Studies have shown that presence of an STI increases risk of transmission of other STIs and increased morbidity [27]. Notably, literature on the effects of STIs on HIV viremia is limited. Condom use was not associated with viral detectability above 200 copies/ml in this analysis.

This study contributes much needed data regarding the factors associated with unsuppressed viremia among African pregnant and breastfeeding women receiving ART treatment for life; more so because viral load testing is only a fairly recent intervention in HIV treatment monitoring. These data demonstrate that socioeconomic barriers and age remain key predictors of viral detectability, as well as recent hospitalization and recent abnormal vaginal discharge.

The strengths in these analyses include that PROMOTE is one of the largest cohort studies of HIV infected women of child bearing age in Africa with a large sample size; and is being conducted in multiple sites in East and Southern Africa, which increases the generalizability of the findings. In addition, there is ongoing quality assurance facilitated by the data management center, and annual research site monitoring as part of the study. Limitations of this study include lack of cART resistance, depression, Intimate Partner Violence (IPV) and stigma data. Furthermore, this is a baseline cross sectional analysis hence virologic failure has not been confirmed. Self-report in response to questionnaires introduces the potential for recall bias. Additionally, this PROMISE-PROMOTE cohort may not be representative of women in the general population.

## Future plans for the study

Follow up trends in viral load and adherence data over the 5 year follow up in PROMOTE will be presented when available; as will the relation of hair drug levels and drug resistance testing correlates. Point- of- care VL testing coupled with motivation adherence counseling and adherence risk assessment tool development are in the process of being implemented at the sites in Zimbabwe and Uganda respectively.

## Conclusion

This baseline analysis of the PROMOTE study set out to evaluate what clinical and socioeconomic factors were associated with a detectable viremia of >200 copies/mL in sub-Saharan African mothers on lifelong cART. Baseline data demonstrate that socioeconomic barriers such as lack of electricity in household premises (a proxy for poverty), not being married, young age, and prior history of missing pill doses remain key predictors of viral detectability. This study suggests that the use of self-reported adherence to cART may still play a role in adherence determination in the absence of superior measures. The most common reasons given by mothers for missing cART doses emphasize the need for effective motivational adherence counseling coupled with economic empowerment, enabling women to improve adherence by using simple reminders and a differentiated care model tailored to mother's needs. The PROMOTE study insights provide opportunities for possible development and improvement of targeted /cost effective implementation strategies to help support lifetime maternal adherence to both cART and HIV care.

## Supporting information

**S1 Table. Manuscript data.**
(XLSX)

**S2 Table. Variable name and label.**
(XLSX)

**S3 Table. Sensitivity analyses using the 1000 copies/ml WHO threshold.**
(DOCX)

## Acknowledgments

We thank the women and children who are participating in the PROMOTE study at each of the research sites. We acknowledge the research teams at each of the following sites: MUJHU, Kampala, Uganda; UNC Project Clinical Research Site, Lilongwe, Malawi; Johns Hopkins-College of Medicine Research Project, Blantyre, Malawi; University of Zimbabwe College of Health Sciences Clinical Trials Research Centre (UZCHS-CTRC) Zimbabwe; Perinatal HIV Research Unit (PHRU), Soweto, South Africa; Centre for the AIDS Programme of Research in South Africa (CAPRISA), uMlazi Clinical Research Site, Durban, South Africa.

The findings and conclusions reported herein are those of the author(s) and do not necessarily reflect the official position of the U.S. government.

## Author Contributions

**Conceptualization:** Patience Atuhaire, Sherika Hanley.

**Data curation:** Patience Atuhaire, Sherika Hanley, Nonhlanhla Yende-Zuma, Jim Aizire, Taha Taha, Mary Glenn Fowler.

**Formal analysis:** Patience Atuhaire, Sherika Hanley, Nonhlanhla Yende-Zuma, Jim Aizire, Taha Taha, Mary Glenn Fowler.

**Funding acquisition:** Taha Taha, Mary Glenn Fowler.

**Investigation:** Patience Atuhaire, Sherika Hanley, Jim Aizire, Taha Taha, Mary Glenn Fowler.

**Methodology:** Patience Atuhaire, Sherika Hanley, Jim Aizire, Mary Glenn Fowler.

**Project administration:** Taha Taha.

**Resources:** Taha Taha, Mary Glenn Fowler.

**Supervision:** Patience Atuhaire, Sherika Hanley, Nonhlanhla Yende-Zuma, Jim Aizire, Lynda Stranix-Chibanda, Bonus Makanani, Beteniko Milala, Haseena Cassim, Taha Taha, Mary Glenn Fowler.

**Writing – original draft:** Patience Atuhaire, Sherika Hanley, Nonhlanhla Yende-Zuma.

**Writing – review & editing:** Patience Atuhaire, Sherika Hanley, Nonhlanhla Yende-Zuma, Jim Aizire, Lynda Stranix-Chibanda, Bonus Makanani, Beteniko Milala, Haseena Cassim, Taha Taha, Mary Glenn Fowler.

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
