## [Decision Letter · Decision Letter 0]

28 Jul 2019

PONE-D-19-17608

Factors associated with unsuppressed viremia in women living with HIV on lifelong ART in a multi-country cohort study: US-PEPFAR PROMOTE study.

PLOS ONE

Dear Dr Atuhaire,

Thank you for submitting your manuscript to PLOS ONE. After careful consideration, we feel that it has merit but does not fully meet PLOS ONE’s publication criteria as it currently stands. Therefore, we invite you to submit a revised version of the manuscript that addresses the points raised during the review process.

We would appreciate receiving your revised manuscript by Sep 11 2019 11:59PM. To enhance the reproducibility of your results, we recommend that if applicable you deposit your laboratory protocols in protocols.io, where a protocol can be assigned its own identifier (DOI) such that it can be cited independently in the future. For instructions see: http://journals.plos.org/plosone/s/submission-guidelines#loc-laboratory-protocols

We look forward to receiving your revised manuscript.

Kind regards,

Marcel Yotebieng

Academic Editor

PLOS ONE

Journal Requirements:

Additional Editor Comments (if provided):

Reviews have made many suggestions as to how to improve the readability of the manuscript and validity of the results. A you revised the manuscript, make to not overstate the application/generilizability of the results.

Reviewers' comments:

Reviewer's Responses to Questions

**Comments to the Author**

1. Is the manuscript technically sound, and do the data support the conclusions?

Reviewer #1: Yes

Reviewer #2: Yes

Reviewer #3: Yes

Reviewer #4: Partly

2. Has the statistical analysis been performed appropriately and rigorously? 

Reviewer #1: Yes

Reviewer #2: Yes

Reviewer #3: Yes

Reviewer #4: No

3. Have the authors made all data underlying the findings in their manuscript fully available?

Reviewer #1: Yes

Reviewer #2: Yes

Reviewer #3: Yes

Reviewer #4: No

4. Is the manuscript presented in an intelligible fashion and written in standard English?

Reviewer #1: Yes

Reviewer #2: Yes

Reviewer #3: No

Reviewer #4: Yes

5. Review Comments to the Author

Reviewer #1: The authors conducted an analysis to explore factors associated with viremia above 200 among women from multiple sites in Africa. I found the paper to be written well with a clear logical flow and tight focus. However, I have identified minor concerns that the authors should work on.

- Page 11, line 87 …are most strongly assoicated with 87 unsuppressed…..

Correct the typo

- Font size is not consistent, see beginning of results section, and probably other places

- Footnote in table2

a2 missing data, b1 missing data, camongst 1549

What do these numbers 2 and 1 supposed to represent?

- Reference 2, 2018 [ Is there something missing, it appears as if you started unclosed bracket?

Reviewer #2: July 21, 2019

Factors associated with unsuppressed viremia in women living with HIV on lifelong ART in a mult-country cohort study: US-PEPFAR PROMOTE Study.

PLOS ONE

Summary: This manuscript describes factors related to viremia in WHLIV in sub-Saharan Africa within the PROMOTE multi-site study. The topic is of interest to clinicians and policy makers focused on caring for both WLHIV and pregnant/postpartum WLHIV. The manuscript is generally well written though does not present much novel data and has several gaps outlined below.

Suggested major revisions:

1. Introduction: Throughout the introduction the authors make assertions that they do not support with sufficient references. This includes Lines 77, 81, 82 and 109.

2. Introduction: description of the PROMISE study “…women randomized… to initiate varied ARV regimens during pregnancy….” Lines 99-103 is not well worded. Presumably women were not randomized in response to the START study though study protocols may have changed after START results were released?

3. Methods: inclusion criteria from PROMISE should be included since that defines the cohort currently being followed.

4. A bias could be introduced by allowing site teams to “judge social or other reasons which would make it difficult for mother/child pair to comply with study requirements” Is there any actual criteria to define this? How many mother/child pairs were excluded due to site judgement decisions? If this is a substantial number/% (>5%) authors should add to limitations.

5. Results: a flow diagram of PROMISE participants who then rolled over to PROMOTE and reasons for exclusion, missing baseline VL etc should be included

6. The pregnancy and postpartum status (e.g. months postpartum) should be reported either at time of VL reported or at least enrollment to PROMOTE to give context to findings.

7. Figure 1 does not add anything to the text presented.

8. If available, ART regimen should be included in Table 1. Since PROMISE randomized to different regimens and regimen is a known factor affecting viremia. Also would be interesting to note if women had changed from their PROMISE regimen.

9. Unclear what “awareness of dosing instructions” means and how this question was asked, given it is a yes/no response.

10. Discussion: Authors confuse the UNAIDS 90-90-90 targets. The third 90 actually represents 90% of women on ART or 73% of all WHLIV (diagnose 90%, put 90% of diagnosed women on ART had achieve viral suppression in 90% of those on ART).

11. Condom use result is randomly added to paragraph on reasons for ARV non-adherence

12. Separate paragraph on differentiated care models could be combined with section on travel and forgetting pills.

13. The statement “one would expect less disclosure issues and united communities taking treatment together” seems judgmental and not supported by literature demonstrating ongoing issues with stigma in high prevalence areas.

14. Authors do not effectively argue that pill burden and low toxicity should reduce barriers to adherence since they do not describe what ART regimens or formulations anywhere in results. Most countries still recommend efavirenz-based regimens for WHLIV which may have substantial side effects for a significant proportion of women.

15. Lines 276-284 seem to jump from result to result without an overlying theme. Why recent vaginal discharge is associated with viremia is not explored at all.

16. Lack of resistance data is a major limitation as the authors argue that the VL presented are all due to adherence. Additionally, lack of data on depression, IPV, stigma are also limitations.

17. Authors should also note bias in sample as all participants are from those retained in a long term study (PROMISE) which is not usually representative of the general population. This may have resulted in lower viremia than WLHIV generally.

18. While the authors found an association between self-reported adherence in a study setting and VL, this is often not the case in routine programmatic settings where patients have a disincentive to truthfully report adherence. This conclusion should be modified.

19. The author suggest motivational interviewing and differentiated care models will be effective but it is unclear how this will impact SES factors, young age, and marital status.

Minor revisions

20. Abstract: the conclusion suggests not being married and young age are socioeconomic barriers- suggest re-wording.

21. Methods: reword “one of the longest ongoing follow up epidemiologic multinational cohorts”

22. Lines 184- 185 repetitive phrasing (Notably, of note)

23. Discussion suggests SES is a known predictor of adherence while the data actually seem mixed on this and the references do not support this assertion.

24. Rephrase “steers away” line 240

Reviewer #3: This is an important manuscript addressing predictors of viremia in women of childbearing age living with HIV across sub-saharan Africa in an observational cohort study. This is generally very well presented. Below I present minor comments. In addition, a close review for typos/grammatical errors would be worthwhile.

Abstract

“Not being married” is mentioned in conclusion but not results. Consider adding to results (or dropping from conclusion) in abstract so the conclusions are drawn follow the results presented.

Intro

line 79

…”satisfaction with health worker information availed were conducive”… —> reword for clarity

Line 82

…”This may be attributed to less motivation to protect the child post delivery” —> perhaps clarify that the motivation may decline when the risk is no longer present (currently sounds as though the motivation declines but not clear that the risk has declined)

Methods - Eligibility

How long was the minimum time on ART prior to the enrollment VL - did you apply any restrictions for inclusion, e.g., at least 3 or 6 months of ART?

RESULTS

Starting Line 198: “The predictors of detectable viremia ….” - clarify you are referring to the multivariable anaysis

Line 196: This sentence:

The most common reason for 197 missing ART dosing was travelling without sufficient ARV supply and simply forgetting (Fig 2).

This is in the middle of a paragraph about Table 2 results and is jarring to jump here and jump back - perhaps reasons deserves its own paragraph

Table 1

= Clarify in the title or footer that these are row percents — that they are not baseline characteristics stratified by VL status

- since you are showing row percents, it would be helpful to include all levels of categorical variables., e.g, for highest level of education we only see completed secondary but would be nice to see the % non suppressed with less edu

Typo, for example -

Line 91-92

Thus the purpose of this analyses —> should be these or analysis

Reviewer #4: Please address the following major issues that need addressed before the paper could be published.

Title

1) Reading the title can be misleading in that it would make a reader think that you are performing a cohort study. Please reformulate to make the study design of the present analysis clear (which I think is cross-sectional), rather than focusing on the parent study.

Abstract

2) The designation “Africa women” may be misleading. It may be too general given that your sample is comprised of women from Eastern and Southern Africa. Countries such as Egypt and Libya are parts of Africa and if you believe that your findings apply to these countries, please elaborate so. However, please reformulate all instances of the word “Africa” to be more specific to your population of interest.

3) Please include the design of your analysis in the abstract.

4) In line 42, did you mean “to assess”

5) Please include the measure of association that you aimed to estimate in the method section.

6) The conclusion reads “Socioeconomic barriers such as poverty, not being married, young age, and self-reported missed doses remain key predictors of unsuppressed viremia.” I’d suggest restructuring the sentence because it seems that “not being married, young age…” are part of the socioeconomic barriers.

7) The word “remain” in the conclusion can be misleading because the present analysis is not longitudinal in nature. If you want to refer to a previous publication of the same group, please elaborate so.

Introduction

8) Line 66, please update the statistics about the number of people receiving cART, to a more recent date. Your statistics are from 2017.

9) Line 70, “ … 90-90-90 2020 Strategy in ending the epidemic by 2030 (2).” Does not read well. There is a capital letter in the middle of sentence.

10) Line 87. Please correct the word “assoicated”

11) Line 90. Suggest not starting “Tuberculosis” with capital letter.

12) Please provide a relevant reference to the following sentence “Emerging antriretroviral drug resistance has been known to occur from levels of 200 copies/ml or above, and use of this threshold eliminates most cases of apparent viremia caused by viral load blips or assay variability”

The WHO considered VL>1000 as threshold to define viral failure, to reduce pitfalls from viral blips. If you believe that VL<200 eliminates most viral blips, please provide the reference supporting this.

Method

13) Please focus on the description of the design of this present study. You were focusing on describing the parent study PROMOTE.

14) Please indicate what was defined as “high enrolling PROMISE sites”

15) Please briefly describe the intervention in PROMISE study, to relieve any concern that the PROMISE intervention could have an impact on viremia among women in your study sample.

16) By definition, exclusion criteria are characteristics of subjects who have met the inclusion criteria. Therefore, if “willing to provide informed consent to enroll and continue follow-up” is an inclusion criterion, it is would be redundant to write “unwilling to provide informed consent to continue follow-up” as exclusion criterion, because both represent the same thing. If you meant to say that participants who refused to provide consent after being enrolled are excluded, I would suggest making it clearer.

17) Please be specific about criteria used to make the judgment in line 132

18) Please add in the method section a description of covariates: how they were defined and coded. In addition, please indicate the justification for each coding scheme. If your group has used the coding in a previous manuscript, please provide relevant citation.

19) Could you add information about the testing procedure: blood specimen manipulation, storage, type of laboratory, existing quality control.

20) I do not think that the following sentence is relevant to your present objectives “Samples were stored for future HIV drug resistance testing (blood) and cumulative drug levels testing (hair).”

21) Please clearly indicate why did you use the Poisson regression and why you did not use log binomial regression.

22) You mentioned that you used prevalence risk ratio as measure of association. However, in the abstract you mentioned that you used prevalence rate ratio. I would suggest being consistent.

23) You wrote that each predictor was fitted in the model as first step of your modeling(line 159). However, in line 155 it reads “Fisher’s exact, Chi-square test of independence or Wilcoxon rank sum tests were used to test for an association between baseline characteristics and unsuppressed viremia”

Which one was used in the univariable analyses ?

24) “and (ii) multivariable model with all the predictors included in the model” This sentence has no verb

25) It seems as if you are contradicting yourselves by including the following phrases in the method section “…. multivariable model with all the predictors included in the model” AND “Variables included in the multivariable analyses were chosen based on prior research on risk factors, biological plausibility, and previously identified clinical associations”. The ALL in the first sentence may be understood as you included all covariates from the univariable analyses in the multivariable model.

26) The following sentence “… prior research on risk factors, biological plausibility, and previously identified clinical associations”

Using “Prior research” and “previously identified clinical associations” in the same sentence is redundant. In addition, please provide citation for the previously identified clinical associations.

Results

27) Please provide in the supplemental materials the results using VL>1000 as secondary outcome.

28) Please include a flowchart or a relevant table that summarizes the following information “Overall, 1987 mothers were enrolled into the PROMOTE study, of whom 1947 (98%) women reported taking ART at the enrolment visit and HIV-1 viral load results were available for 1934. HIV-1 VL was above the limit of quantification in 293/1934 (15 %). ”

29) Table 1 does not provide with an overall description of the sample, because it is stratified by viral suppression status. Please add a third column containing an overall frequency.

30) The notations p-values are not consistent, sometimes they are written as “<.001 ” . Sometimes, they are written as “<0.001”

31) For each association you describe in the result, please include the comparison group.

32) Please be consistent in the notation of copies/ml. Sometimes you use cp/ml, sometimes you use copies/ml

33) “In addition, the absence of socioeconomic factors such as electricity in the premises” The sentence does not read well. Which premises ?

34) Did you mean “travel time from the cART clinic “ or “travel time to the cART clinic “

35) I do not think that the following sentence is relevant with regards to your study objective. “Additionally, about 10% (n=29) of women with unsuppressed viremia had an HIV-uninfected partner.”

Discussion

36) The following sentence does not read well. “We identify sociodemographic, self-reported non-adherence, and clinical factors that were associated with detectable viremia > 200copies/ml. ”

37) LINE 230, you used “correlation” …. Correlation analyses and regression analyses are very different.

38) Line 292: The longitudinal design of the PROMOTE does not confer any strength to this present analysis, which is a cross-sectional analysis of the baseline data of a longitudinal analysis. However, the sample size is a strength. Please reformulate.

39) Please elaborate more on the ongoing quality assurance that would strengthen this present cross-sectional analysis.

40) There are additional limitations that ought to be discussed: potential recall bias due to self-report, the inability to confirm viral failure.

6. PLOS authors have the option to publish the peer review history of their article (what does this mean?). If published, this will include your full peer review and any attached files.

Reviewer #1: No

Reviewer #2: No

Reviewer #3: No

Reviewer #4: No

---

## [Author Response · Author response to Decision Letter 0]

11 Sep 2019

Comments from Reviewer 1

The authors conducted an analysis to explore factors associated with viremia above 200 among women from multiple sites in Africa. I found the paper to be written well with a clear logical flow and tight focus. However, I have identified minor concerns that the authors should work on.

Comment: Page 11, line 87 …are most strongly associated with 87 unsuppressed…..

Response: Thank you for this comment. Correction made.

Comment: Font size is not consistent, see beginning of results section, and probably other places

Response: Font size has been revised

Comment: Footnote in table2 a2 missing data, b1 missing data, camongst 1549

What do these numbers 2 and 1 supposed to represent?

Response: These represent the number of participants with missing data for a specific variable. We have re-worded the footnote clearly to indicate that.

Comment: Reference 2, 2018 [ Is there something missing, it appears as if you started unclosed bracket?

Response: Bracket removed

Comments from Reviewer 2

Factors associated with unsuppressed viremia in women living with HIV on lifelong ART in a multi-country cohort study: US-PEPFAR PROMOTE Study.

PLOS ONE

Summary: This manuscript describes factors related to viremia in WHLIV in sub-Saharan Africa within the PROMOTE multi-site study. The topic is of interest to clinicians and policy makers focused on caring for both WLHIV and pregnant/postpartum WLHIV. The manuscript is generally well written though does not present much novel data and has several gaps outlined below.

Suggested major revisions:

1. Introduction: Throughout the introduction the authors make assertions that they do not support with sufficient references. This includes Lines 77, 81, 82 and 109.

Response: Thank you for raising this. Additional references have been added.

2. Introduction: description of the PROMISE study “…women randomized… to initiate varied ARV regimens during pregnancy….” Lines 99-103 is not well worded. Presumably women were not randomized in response to the START study though study protocols may have changed after START results were released?

Response: Thank you for this comment. The sentence has been reworded to read better. The PEPFAR-funded PROMOTE study presents a unique opportunity to assess longer term treatment outcomes among women previously randomized to different antiretroviral (ARV) regimens during pregnancy and postpartum for the purpose of preventing perinatal HIV transmission in the PROMISE trial. Subsequently, all women transitioned to lifelong cART in response to the START study which showed clear benefit of universal ART in June 2015

3. Methods: inclusion criteria from PROMISE should be included since that defines the cohort currently being followed.

Response: Thank you for this comment. Inclusion criteria for PROMISE has been added.

4. A bias could be introduced by allowing site teams to “judge social or other reasons which would make it difficult for mother/child pair to comply with study requirements” Is there any actual criteria to define this? How many mother/child pairs were excluded due to site judgement decisions? If this is a substantial number/% (>5%) authors should add to limitations.

Response: Thank you for noting this. The criteria included those women who felt that continued follow up may predispose them to social harm and women who planned on relocating home out of the study catchment area and could not keep up with study visit requirements afterwards. We have added a table to indicate numbers from PROMISE study and the lost to follow up.

5. Results: a flow diagram of PROMISE participants who then rolled over to PROMOTE and reasons for exclusion, missing baseline VL etc should be included

Response: Thank you for this comment. We have included a table to show the numbers from PROMISE to PROMOTE including those that were lost to follow up at PROMOTE study entry. The reasons for non-enrolment were declined participation, some women felt that continued follow up may predispose them to social harm and were excluded, women who had relocated out of the study catchment area and could not keep up with study visit requirements afterwards were also excluded. 

6. The pregnancy and postpartum status (e.g. months postpartum) should be reported either at time of VL reported or at least enrollment to PROMOTE to give context to findings.

Response: thank you for raising this. Additional text to indicate that enrolled women were either having a subsequent pregnancy or not pregnant (up to 4 years post-delivery in PROMISE) or breastfeeding babies born between PROMISE study close up and PROMOTE study start has been added. This information is also in Table 2.

7. Figure 1 does not add anything to the text presented.

Response: Figure 1 has been deleted.

8. If available, ART regimen should be included in Table 1. Since PROMISE randomized to different regimens and regimen is a known factor affecting viremia. Also would be interesting to note if women had changed from their PROMISE regimen.

Response: The summary of the regimens is shown in Table 1. Most women were switched from the PROMISE regimen (TDF/FTC/LPV/r) to TDF/3TC/EFV (standard of care regimen).

9. Unclear what “awareness of dosing instructions” means and how this question was asked, given it is a yes/no response.

Response: The question re-awareness of dosing instructions was phrased as “ Do you know if the ARV medications you are taking need to be taken on a schedule, such as “once a day” or “2 times a day” or “every 8 hours”? Yes/No/Don’t know.

10. Discussion: Authors confuse the UNAIDS 90-90-90 targets. The third 90 actually represents 90% of women on ART or 73% of all WHLIV (diagnose 90%, put 90% of diagnosed women on ART had achieve viral suppression in 90% of those on ART).

Response: Thank you for raising this. We have rephrased to state results in line with the UNAIDS 90% target.

11. Condom use result is randomly added to paragraph on reasons for ARV non-adherence

Response: This has been modified and condom use result has been added under the paragraph on vaginal discharge. 

12. Separate paragraph on differentiated care models could be combined with section on travel and forgetting pills.

Response: Thank you for this suggestion. We have combined the section on travel with that of forgetting pills.

13. The statement “one would expect less disclosure issues and united communities taking treatment together” seems judgmental and not supported by literature demonstrating ongoing issues with stigma in high prevalence areas.

Response: Thank you for raising this important issue. We have deleted this statement.

14. Authors do not effectively argue that pill burden and low toxicity should reduce barriers to adherence since they do not describe what ART regimens or formulations anywhere in results. Most countries still recommend efavirenz-based regimens for WHLIV which may have substantial side effects for a significant proportion of women.

Response: Thank you for this query. We have provided data on ART regimens. About 88% of the women were on Efavirenz or Nevirapine based regimens. 

15. Lines 276-284 seem to jump from result to result without an overlying theme. Why recent vaginal discharge is associated with viremia is not explored at all.

Response: This entire paragraph has been revised.

16. Lack of resistance data is a major limitation as the authors argue that the VL presented are all due to adherence. Additionally, lack of data on depression, IPV, stigma are also limitations.

Response: These limitations have been added to the study limitation section.

17. Authors should also note bias in sample as all participants are from those retained in a long term study (PROMISE) which is not usually representative of the general population. This may have resulted in lower viremia than WLHIV generally.

Response: Thank you for this comment. We have added this information as part of the limitations.

18. While the authors found an association between self-reported adherence in a study setting and VL, this is often not the case in routine programmatic settings where patients have a disincentive to truthfully report adherence. This conclusion should be modified.

Response: Thank you for this comment. The conclusion has been reworded to read “…. the use of self-reported adherence to cART may still play a role in adherence determination in the absence of superior measures.”

19. The author suggest motivational interviewing and differentiated care models will be effective but it is unclear how this will impact SES factors, young age, and marital status.

Response: Thank you for this comment. We have reworded to include the need for economic empowerment in addition to the motivational interviewing under the conclusion.

Minor revisions

20. Abstract: the conclusion suggests not being married and young age are socioeconomic barriers- suggest re-wording.

Response: Conclusion has been reworded.

21. Methods: reword “one of the longest ongoing follow up epidemiologic multinational cohorts”

Response: The methods section has been reworded.

22. Lines 184- 185 repetitive phrasing (Notably, of note)

Response: Thank you for noting this. This has been modified accordingly. 

23. Discussion suggests SES is a known predictor of adherence while the data actually seem mixed on this and the references do not support this assertion.

Response: Additional references have been added.

24. Rephrase “steers away” line 240

Response: This has been rephrased.

Comments from Reviewer 3

Comment: This is an important manuscript addressing predictors of viremia in women of childbearing age living with HIV across sub-saharan Africa in an observational cohort study. This is generally very well presented. Below I present minor comments. In addition, a close review for typos/grammatical errors would be worthwhile.

Response: Thank you for this. A close review has been done

Abstract

Comment: “Not being married” is mentioned in conclusion but not results. Consider adding to results (or dropping from conclusion) in abstract so the conclusions are drawn follow the results presented.

Response: Thank you for picking this, it was an oversight. This has been rectified and marital status has been included in the results section. We would like to make a note that even though the p-value for marital status is greater than 0.05 but we felt that the 32% increased risk of detectable viremia for those not married is an important finding.

Comment : Intro line 79…”satisfaction with health worker information availed were conducive”… —> reword for clarity

Response: This sentence has been reworded for more clarity.

Comment: Line 82

…”This may be attributed to less motivation to protect the child post-delivery” —> perhaps clarify that the motivation may decline when the risk is no longer present (currently sounds as though the motivation declines but not clear that the risk has declined)

Response: Thank you for this comment. We have reworded for more clarity.

Methods – Eligibility

Comment: How long was the minimum time on ART prior to the enrollment VL - did you apply any restrictions for inclusion, e.g., at least 3 or 6 months of ART?

Response: The range time on ART to enrolment was 1 to 5 years. We did not apply any restrictions

RESULTS

Comment: Starting Line 198: “The predictors of detectable viremia ….” - clarify you are referring to the multivariable analysis

Response: The sentence has been revised to “Results from the multivariable model are shown in Table 2”

Comment: Line 196: This sentence:

The most common reason for 197 missing ART dosing was travelling without sufficient ARV supply and simply forgetting (Fig 2).

This is in the middle of a paragraph about Table 2 results and is jarring to jump here and jump back - perhaps reasons deserves its own paragraph

Response: A new paragraph has been created as recommended. 

Table 1

Comment: = Clarify in the title or footer that these are row percents — that they are not baseline characteristics stratified by VL status

Response: The following sentence has been added in the footer of Table 1. “Please note that we presented row percentages to display the proportion of patients with detectable viremia for each level of the variable.”

Comment: since you are showing row percents, it would be helpful to include all levels of categorical variables., e.g, for highest level of education we only see completed secondary but would be nice to see the % non-suppressed with less education

Response: This has been inserted in Table 1 for other variables as well.

Comment: Typo, for example -Line 91-92

Thus the purpose of this analyses —> should be these or analysis

Response: This has been changed as suggested

Comment from Reviewer 4

Please address the following major issues that need addressed before the paper could be published.

Title

1) Reading the title can be misleading in that it would make a reader think that you are performing a cohort study. Please reformulate to make the study design of the present analysis clear (which I think is cross-sectional), rather than focusing on the parent study. 

Response: Thank you for bringing this to our attention. The title has been amended accordingly to “Factors associated with unsuppressed viremia in women living with HIV on lifelong ART in the multi-country US-PEPFAR PROMOTE study: A cross-sectional analysis.”

Abstract 

2) The designation “Africa women” may be misleading. It may be too general given that your sample is comprised of women from Eastern and Southern Africa. Countries such as Egypt and Libya are parts of Africa and if you believe that your findings apply to these countries, please elaborate so. However, please reformulate all instances of the word “Africa” to be more specific to your population of interest.

 Response: The authors have further clarified Africa to sub-Saharan Africa.

3) Please include the design of your analysis in the abstract. 

Response: The study design has been included in the abstract as follows. “This cross-sectional analysis was based on baseline data of the PROMOTE longitudinal cohort study at 8 sites in Uganda, Malawi, Zimbabwe and South Africa”

4) In line 42, did you mean “to assess”.

Response: Yes, however this sentence has been deleted.

5) Please include the measure of association that you aimed to estimate in the method section. 

Response: This was already described under the statistical analyses section. We used prevalence risk ratios to measure the strength of an association between baseline characteristics and unsuppressed viremia.

6) The conclusion reads “Socioeconomic barriers such as poverty, not being married, young age, and self-reported missed doses remain key predictors of unsuppressed viremia.” I’d suggest restructuring the sentence because it seems that “not being married, young age…” are part of the socioeconomic barriers.

Response: The sentence has been restructured in lines 60-61 “Socioeconomic barriers such as poverty, and individual barriers like not being married, young age, and self-reported missed doses, are key predictors of unsuppressed viremia.”

7) The word “remain” in the conclusion can be misleading because the present analysis is not longitudinal in nature. If you want to refer to a previous publication of the same group, please elaborate so. 

Response: The sentence has been restructured in lines 60-61 as indicated above.

Introduction 

8) Line 66, please update the statistics about the number of people receiving cART, to a more recent date. Your statistics are from 2017. 

Response: Thank you for highlighting this. Statistics have been updated in line 73-74. “Subsequently with the introduction of the Universal “Test and Treat” strategy, approximately 23.3 million people (including women) had access to combination Antiretroviral Therapy (cART) globally in 2018.” 

9) Line 70, “ … 90-90-90 2020 Strategy in ending the epidemic by 2030 (2).” Does not read well. There is a capital letter in the middle of sentence.

Response: This has been corrected in line 76.

10) Line 87. Please correct the word “assoicated”. 

Response: Correction has been made in line 92.

11) Line 90. Suggest not starting “Tuberculosis” with capital letter. 

Response: Correction has been done in line 95.

12) Please provide a relevant reference to the following sentence “Emerging antriretroviral drug resistance has been known to occur from levels of 200 copies/ml or above, and use of this threshold eliminates most cases of apparent viremia caused by viral load blips or assay variability”

The WHO considered VL>1000 as threshold to define viral failure, to reduce pitfalls from viral blips. If you believe that VL<200 eliminates most viral blips, please provide the reference supporting this. 

Response: Thank you for noting this. 2 references (14, 15) have been included.

Method 

13) Please focus on the description of the design of this present study. You were focusing on describing the parent study PROMOTE.

Response: The present study design has been clarified in line 120.

14) Please indicate what was defined as “high enrolling PROMISE sites”. 

Response: This sentence has been removed since it is not relevant to this analysis. However, high enrolling sites were those sites that had more numbers of enrollments by the time accrual into the PROMISE trial had closed.

15) Please briefly describe the intervention in PROMISE study, to relieve any concern that the PROMISE intervention could have an impact on viremia among women in your study sample. 

Response: Briefly described in lines 104-108 and 128-131. It reads ‘this cohort presents a unique opportunity to assess longer term treatment outcomes among women previously randomized to different antiretroviral (ARV) regimens (WHO option A and option B) during pregnancy and postpartum for the purpose of preventing perinatal HIV transmission. Lines 128-131 describes the regimens most women were receiving in each country.

16) By definition, exclusion criteria are characteristics of subjects who have met the inclusion criteria. Therefore, if “willing to provide informed consent to enroll and continue follow-up” is an inclusion criterion, it is would be redundant to write “unwilling to provide informed consent to continue follow-up” as exclusion criterion, because both represent the same thing. If you meant to say that participants who refused to provide consent after being enrolled are excluded, I would suggest making it clearer.

Response: Exclusion criteria have been restructured in lines 138-142.

17) Please be specific about criteria used to make the judgment in line 132. 

Response: An example has been provided in lines 138-142.

18) Please add in the method section a description of covariates: how they were defined and coded. In addition, please indicate the justification for each coding scheme. If your group has used the coding in a previous manuscript, please provide relevant citation.

Response: There was no coding scheme used. Guided by the study’s schedule of evaluations, covariates were collected using baseline questionnaires and laboratory procedures at study entry, as indicated in lines 149-156. In Table 2, most variables were taken directly from the questionnaire and laboratory results. The levels of the variables that are combined (e.g. marital status, travel time to the cART clinic, missed ART doses) are provided in Table 1.

19) Could you add information about the testing procedure: blood specimen manipulation, storage, type of laboratory, existing quality control. 

Response: In lines 150-153“Assays were done at various Good Clinical Laboratory Practice (GCLP) and Clinical Laboratory Improvement Amendments (CLIA) compliant site laboratories adherent to standards for proper collection, processing, labeling, and transport of specimens” has been added.

20) I do not think that the following sentence is relevant to your present objectives “Samples were stored for future HIV drug resistance testing (blood) and cumulative drug levels testing (hair).” 

Response: This sentence has been removed.

21) Please clearly indicate why did you use the Poisson regression and why you did not use log binomial regression.

Response: The log binomial model failed to converge as more variables were added into the multivariable model. Therefore, we opted for Poisson regression with robust variance.

22) You mentioned that you used prevalence risk ratio as measure of association. However, in the abstract you mentioned that you used prevalence rate ratio. I would suggest being consistent. 

Response: We have changed rate ratio to risk ratio

23) You wrote that each predictor was fitted in the model as first step of your modeling(line 159). However, in line 155 it reads “Fisher’s exact, Chi-square test of independence or Wilcoxon rank sum tests were used to test for an association between baseline characteristics and unsuppressed viremia”

Which one was used in the univariable analyses? 

Response: We used to Fisher’s exact, Chi-square test of independence or Wilcoxon rank sum tests to test for an association between baseline characteristics and unsuppressed viremia. However, to determine the strength of the association and also to identify variables that are predictive of unsuppressed viremia, we used both univariable and multivariable Poisson models.

24) “and (ii) multivariable model with all the predictors included in the model” This sentence has no verb

Response: The sentence has been revised and it now reads: “and (ii) multivariable model with multiple predictors included in the model”

25) It seems as if you are contradicting yourselves by including the following phrases in the method section “…. multivariable model with all the predictors included in the model” AND “Variables included in the multivariable analyses were chosen based on prior research on risk factors, biological plausibility, and previously identified clinical associations”. The ALL in the first sentence may be understood as you included all covariates from the univariable analyses in the multivariable model.

Response: Thank you. The sentences have been fixed.

26) The following sentence “… prior research on risk factors, biological plausibility, and previously identified clinical associations” Using “Prior research” and “previously identified clinical associations” in the same sentence is redundant. In addition, please provide citation for the previously identified clinical associations.

Response: The sentence has been reformulated and citations have been included in the introduction (line 93, 96).

Results

27) Please provide in the supplemental materials the results using VL>1000 as secondary outcome.

Response: There were 166/1934 participants with VL>1000 and results are presented in Supplementary Tables 1 and 2. The following sentence was also inserted under the statistical analyses section. ‘Furthermore, in accordance with WHO threshold for treatment failure, we performed sensitivity analyses and defined unsuppressed viremia as viral load above 1000 copies/ml. “

28) Please include a flowchart or a relevant table that summarizes the following information “Overall, 1987 mothers were enrolled into the PROMOTE study, of whom 1947 (98%) women reported taking ART at the enrolment visit and HIV-1 viral load results were available for 1934. HIV-1 VL was above the limit of quantification in 293/1934 (15 %). ”

Response: A flow chart (figure 1) has been created to summarize the above information.

29) Table 1 does not provide with an overall description of the sample, because it is stratified by viral suppression status. Please add a third column containing an overall frequency. 

Response: Table 1 has been amended to show total numbers

30) The notations p-values are not consistent, sometimes they are written as “<.001”. Sometimes, they are written as “<0.001”

Response: Thank you, this has been fixed

31) For each association you describe in the result, please include the comparison group.

Response: The results section has been revised accordingly.

32) Please be consistent in the notation of copies/ml. Sometimes you use cp/ml, sometimes you use copies/ml

Response: Thank you. Corrections have been made.

33) “In addition, the absence of socioeconomic factors such as electricity in the premises” The sentence does not read well. Which premises? 

Response: This has been further clarified to household premises.

34) Did you mean “travel time from the cART clinic “ or “travel time to the cART clinic “

Response: It is travel time to the cART clinic. Corrections made.

35) I do not think that the following sentence is relevant with regards to your study objective. “Additionally, about 10% (n=29) of women with unsuppressed viremia had an HIV-uninfected partner.”

 Response: This sentence has been removed.

Discussion

36) The following sentence does not read well. “We identify sociodemographic, self-reported non-adherence, and clinical factors that were associated with detectable viremia > 200copies/ml.”

Response: Thank you for noting this. The sentence has been revised to reads “Among the women with detectable viremia, we identified that socio demographic, self-reported non-adherence, and clinical factors were associated with detectable viremia > 200copies/ml.”

37) LINE 230, you used “correlation” …. Correlation analyses and regression analyses are very different.

Response: The sentence has been revised.

38) Line 292: The longitudinal design of the PROMOTE does not confer any strength to this present analysis, which is a cross-sectional analysis of the baseline data of a longitudinal analysis. However, the sample size is a strength. Please reformulate.

Response: Line 317-318 –Longitudinal design removed.

39) Please elaborate more on the ongoing quality assurance that would strengthen this present cross-sectional analysis.

Response: Quality assurance has been elaborated in lines 150-153 and 320-321.

40) There are additional limitations that ought to be discussed: potential recall bias due to self-report, the inability to confirm viral failure.

Response: Additional limitations have been included.

---

## [Decision Letter · Decision Letter 1]

11 Oct 2019

Factors associated with unsuppressed viremia in women living with HIV on lifelong ART in the multi-country US-PEPFAR PROMOTE study: A cross-sectional analysis

PONE-D-19-17608R1

Dear Dr. Atuhaire,

We are pleased to inform you that your manuscript has been judged scientifically suitable for publication and will be formally accepted for publication once it complies with all outstanding technical requirements.

With kind regards,

Marcel Yotebieng, M.D., MPH, Ph.D

Academic Editor

PLOS ONE

Additional Editor Comments (optional):

Reviewers' comments:

Reviewer's Responses to Questions

**Comments to the Author**

1. If the authors have adequately addressed your comments raised in a previous round of review and you feel that this manuscript is now acceptable for publication, you may indicate that here to bypass the “Comments to the Author” section, enter your conflict of interest statement in the “Confidential to Editor” section, and submit your "Accept" recommendation.

Reviewer #3: All comments have been addressed

Reviewer #4: All comments have been addressed

2. Is the manuscript technically sound, and do the data support the conclusions?

Reviewer #3: Yes

Reviewer #4: Yes

3. Has the statistical analysis been performed appropriately and rigorously? 

Reviewer #3: Yes

Reviewer #4: Yes

4. Have the authors made all data underlying the findings in their manuscript fully available?

Reviewer #3: Yes

Reviewer #4: No

5. Is the manuscript presented in an intelligible fashion and written in standard English?

Reviewer #3: Yes

Reviewer #4: Yes

6. Review Comments to the Author

Reviewer #3: This is an excellent and important manuscript. It is much improved from the original submission and I have no further concerns.

Reviewer #4: (No Response)

7. PLOS authors have the option to publish the peer review history of their article (what does this mean?). If published, this will include your full peer review and any attached files.

Reviewer #3: No

Reviewer #4: No

---

## [Editor Report · Acceptance letter]

17 Oct 2019

PONE-D-19-17608R1 

Factors associated with unsuppressed viremia in women living with HIV on lifelong ART in the multi-country US-PEPFAR PROMOTE study: A cross-sectional analysis 

Dear Dr. Atuhaire:

I am pleased to inform you that your manuscript has been deemed suitable for publication in PLOS ONE. Congratulations! Your manuscript is now with our production department. 

With kind regards,

on behalf of

Dr. Marcel Yotebieng 

Academic Editor

PLOS ONE